# Metabolic Signaling into Chromatin Modifications in the Regulation of Gene Expression

**DOI:** 10.3390/ijms19124108

**Published:** 2018-12-18

**Authors:** Tian Gao, Zyanya Díaz-Hirashi, Francisco Verdeguer

**Affiliations:** Department of Molecular Mechanisms of Disease, University of Zurich, 8057 Zurich, Switzerland; tian.gao@dmmd.uzh.ch (T.G.); zyanya.diaz@dmmd.uzh.ch (Z.D.-H.)

**Keywords:** epigenetics, metabolic signaling, chromatin, obesity

## Abstract

The regulation of cellular metabolism is coordinated through a tissue cross-talk by hormonal control. This leads to the establishment of specific transcriptional gene programs which adapt to environmental stimuli. On the other hand, recent advances suggest that metabolic pathways could directly signal into chromatin modifications and impact on specific gene programs. The key metabolites acetyl-CoA or S-adenosyl-methionine (SAM) are examples of important metabolic hubs which play in addition a role in chromatin acetylation and methylation. In this review, we will discuss how intermediary metabolism impacts on transcription regulation and the epigenome with a particular focus in metabolic disorders.

## 1. Introduction

An essential adaptation of most living organisms is to sense and respond to nutrient availability. A classic example of this interaction is how bacteria regulate expression of metabolic genes in function of the presence of specific nutrients in the environment. These processes were discovered by Nobel laureates Jacob and Monod in the sixties [1]. Mammals have evolved sophisticated ways to respond to nutrients which involve multiple processes and adaptive mechanisms including hormonal inter-organ communication, fasting/feeding and circadian cycles, storage of energy and the possibility to alternate fuel usage. Despite the profound differences between unicellular and pluricellular life in respect to metabolic regulation, it is tempting to speculate that some basic processes could have been conserved or converged during evolution. At the cellular level, multiple metabolic pathways and metabolites are intricately connected to the regulation of gene expression. A paradigmatic example of such regulation is the link of acetyl-CoA between multiple metabolic pathways and its function as donor of acetyl groups for histone acetylation, a crucial chromatin modification involved in active gene transcription.

In this review, we aim at studying how different metabolites impact on gene regulation by directly playing a donor or cofactor function of key chromatin modifying enzymes. In addition, we focus on how these processes could be regulated in different physiological contexts particularly in metabolic disorders. The interaction with diets and potential therapeutic opportunities will also be discussed.

## 2. Interaction of Metabolites with Chromatin Modifications

### 2.1. Acetyl-CoA and Regulation of Acetylation

Acetyl-CoA is a central two-carbon carrier which is produced by different fuel and intermediary sources and connects different metabolic pathways. Carbohydrate catabolism can lead to acetyl-CoA production by the pyruvate dehydrogenase complex (PDH) in the mitochondria, which then feeds the TCA cycle for energy production. The slicing of fatty acids through beta-oxidation produces acetyl-CoA which also enters TCA cycle. In addition, protein degradation can also feed TCA cycle through acetyl-CoA production from ketogenic amino acids. Acetyl-CoA has versatile functions in different crucial pathways. For example, lipids, cholesterol and steroid synthesis are derived from acetyl-CoA utilization as a carbon source. Moreover, ketone bodies are derived from acetyl-CoA. It also plays an important role in processes such as protein glycosylation which will not be discussed in detail in this review.

One of the most abundant chromatin modifications that directly regulates transcriptional activity is histone acetylation [2]. The fact that acetyl-CoA is the only donor for histone acetylation connects the cellular metabolic status with gene control [3]. Generally, when energy status is high, acetyl-CoA levels are elevated and this correlates with global histone acetylation [4,5]. On the contrary, when acetyl-CoA drops, histone acetylation decreases [4,5]. It is not fully understood whether histone acetylation uses particular subcellular pools of acetyl-CoA. Mitochondria contain a large pool of acetyl-CoA, however, its membrane is impermeable for acetyl-CoA. Mitochondrial-produced acetyl-CoA can contribute to cytosolic acetyl-CoA thanks to a shuttle system mediated by ATP Citrate Lyase (ACLY) in the cytoplasm. After acetyl-CoA enters the TCA cycle, citrate is made and transported out to the cytoplasm from the mitochondria (Figure 1). ACLY then catalyzes the conversion of citrate to oxaloacetate by transferring 2 carbons to Coenzyme A leading to acetyl-CoA formation. Wellen et al. elegantly showed that the increase of histone acetylation in response to growth factors and glucose availability is dependent on ACLY [6].

Although the correlation between high energetic status (reflected by high levels of cellular acetyl-CoA) and histone acetylation is well accepted [7], its relevance in specific biological contexts or tissues is not well understood. Whether for example fluctuations of acetyl-CoA levels could lead to differential acetylation on specific loci is not known. In this respect, the activity of Histone Acetyltransferases (HATs) could play a major role. Since the discovery of the first HAT, Gcn5 in yeast, [8] a large number of studies in different fields have identified multiple regulatory functions of HATs not only for histone but also for non-histone protein acetylation [9]. HATs play an essential role in transcriptional activation by forming protein complexes at gene promoters and catalyzing lysine histone acetylation. However, contrary to phosphorylation, the kinetics of histone acetylation largely depends on the acetyl-CoA concentration, particularly acetyl-CoA versus free CoA ratio [3]. While kinases function with saturating ATP concentrations, HATs activity depends on acetyl-CoA concentration. This concentration ranges between 3–30 μM in both yeast and mammalian cells [7] and most HATs have a dissociation constant (Kd) in the low micromolar or high nanomolar range. Therefore, the abundance of acetyl-CoA influences HAT activity, further suggesting a key connection between metabolic status and gene regulation. In addition to concentration, the subcellular localization of acetyl-CoA plays a regulatory role since it is not permeable to mitochondria. This fact leads to the existence of differentially regulated nuclear versus cytosolic acetyl-CoA pools that may indirectly impact HAT activity. It is not known whether additional differential compartmentalization or fluxes of acetyl-CoA could exist within cytosolic and nuclear subcellular compartments. Although acetyl-CoA is freely diffusible between cytoplasm and nucleus, there might be particular physical constraints due to for example high level of chromatin compaction in some nuclear areas which could lead to acetyl-CoA subnuclear compartmentalization [7]. A similar phenomenon has been recently identified for NAD^+^, which has been found to be synthesized differently in subnuclear compartments and concomitantly regulate gene transcription [10]. ACLY is also localized in the nucleus, however, it is not known how its nuclear import is regulated. Recently, DNA damage has been shown to induce ACLY phosphorylation which leads to the recruitment of BRCA1 for homologous recombination in DNA repair [11]. Hence, the translocalization of ACLY may be regulated through post-translational modifications.

### 2.2. NAD^+^ and Regulation of Sirtuin-Dependent Deacetylation

The counterpart of histone acetylation is the removal of acetyl groups by lysine deacetylases (KDACs). These enzymes are divided in two large families: the zinc-dependent deacetylases classically known as histone deacetylases (HDACs), and the family of nicotinamide dinucleotide (NAD^+^) dependent deacetylases; the sirtuins [12,13,14,15]. HDACs usually associate with co-repressors complexes including SIN3A, SMRT or NCOR [15]. Among their wide variety of functions, they also regulate systemic metabolism by the deacetylation of key metabolic transcription factors [16]. There are 7 sirtuins (Sirtuin 1 to 7) that also have non-histone targets. These include transcription factors and metabolic enzymes. The discovery of Sir2 in yeast as a regulator of lifespan extension during caloric restriction conditions exemplified the role of sirtuins in metabolic homeostasis [12]. The key connection of sirtuin’s function with intermediary metabolism is its dependence on NAD^+^. The catalytic deacetylation of substrates by sirtuins uses NAD^+^ as the acceptor of the acetyl group. This reaction leads to the dissociation of NAD^+^ into nicotinamide (NAM) and 2′OAADPr. NAD^+^ is a small molecule ubiquitously present in energy metabolism that serves as a coenzyme of oxidoreductase enzymes in the transfer of electrons between metabolites and NADH. Because the TCA cycle and the electron transport chain (ETC) require both NAD^+^ and NADH, an optimal NAD^+^/NADH ratio is required for an efficient metabolism. The activity of sirtuins consumes NAD^+^ which reduces its concentration and needs to be replenished for basal metabolic functions. Another family of enzymes which consume NAD^+^ are poly-ADP-ribose polymerases (PARPs), which are involved in DNA repair [17]. NAD^+^ in mammals is synthesized from NAM, tryptophan and nicotinic acid. Diet can influence NAD^+^ levels through the uptake of the essential amino acid tryptophan (followed by de novo biosynthesis of NAD^+^), nicotinic acid (Preiss-Handler pathway) or nicotinamide riboside (another precursor of NAM) [18]. The major source of NAD^+^ is NAM through the salvage pathway, however, this view has been recently challenged by recent work showing that tryptophan contributes largely to NAD^+^ synthesis in the liver [19,20]. In the salvage pathway, nicotinamide phosphoribosyltransferase (NAMPT) catalyzes the conversion of NAM into NMN, which is then converted into NAD^+^ by NMNAT. Three different enzymes NMANT1, 2 and 3 catalyze this reaction in nucleus, cytoplasm and mitochondria respectively, where these enzymes are located [10].

### 2.3. Methionine and Regulation of Methylation

Methylation reactions including histone and DNA methylation require methyl groups contributed by dietary methyl donors and by 1-carbon methyl cofactors. The metabolic cycle of methionine is essential to feed most of methylation reactions. Methionine is a sulfur-containing essential amino acid [21] which serves as the substrate for the production of other amino acids such as cysteine and homocysteine [22]. In the methionine metabolic cycle, methionine is firstly converted into S-adenosylmethionine (SAM) by methionine adenosyltransferase (MAT) with the addition of adenosine that is donated by adenosine triphosphate (ATP) [23]. As a universal cellular methyl donor, SAM can be used in different forms of methylation such as DNA, RNA and protein methylation, catalyzed by various methyltransferases. The transfer of the methyl group to the respective substrates leads to the formation of S-adenosylhomocysteine (SAH), which then undergoes hydrolysis into homocysteine by SAH hydrolase with the removal of the adenosine group. Homocysteine can subsequently enter different pathways including remethylation to form back methionine or irreversible transsulfuration for cysteine or α-ketobutyrate production. The catalytic activity of methyltransferases depends interestingly on a high SAM/SAH ratio [14].

In the context of DNA methylation, which typically acts to repress gene transcription, the palindromic CpG dinucleotides in the genome are often the targets for DNA methyltransferases (DNMTs). DNMTs catalyze the transfer of a methyl group from SAM to covalently bind to the carbon-5 position of the cytosine residues in the CpG. This added methyl group can then block transcription factors from binding to the target genome sites, resulting in differential gene regulation [24]. Since methyl groups involved in DNA methylation derived from methionine, an essential amino acid, it suggests that nutrition and metabolism could have an indirect effect on DNA methylation [3]. In the methionine metabolic cycle, it was found that the accumulation of adenosine and homocysteine favors the biosynthesis of SAH rather than hydrolysis [25]. In a study which tested rat liver, it was found that high levels of adenosine and homocysteine could lead to drastic decrease in the ratio of SAM/SAH, resulting in the inhibition of SAM-dependent methyltransferases and a global decrease in DNA methylation [26,27]. Moreover, anti-diabetic drug metformin has recently been found to boost DNA methylation by simultaneously promoting the accumulation of SAM and reducing SAH levels, further suggesting that high SAM:SAH ratio favors DNA methylation [28].

### 2.4. α-Ketoglutarate (αKG) and Regulation of Demethylation

DNA methylation is thought to be a relatively more stable epigenetic marker in comparison to histone modification [29]. Yet, in the past decades, studies have shown that DNA methylation is a dynamic process in different biological contexts [30,31]. There are in general two types of DNA demethylation: active or passive. Active DNA methylation is believed to be an enzyme-catalyzed process which leads to the removal or modification of the methyl group from 5-methylcytosine (5mC) which is mostly present in the CpG dinucleotides. Passive DNA demethylation on the other hand, refers to the loss of methyl group from 5mC during rounds of replication when DNMTs activities are reduced or inhibited. Although the passive mechanism is generally well studied, the active DNA demethylation mechanism still remains to be elucidated [30,32]. There are currently several proposed mechanisms for active DNA demethylation, one of them is oxidative demethylation which links cellular metabolism with DNA demethylation.

Oxidative demethylation is dependent on a group of enzymes called ten eleven translocation (TET) proteins that belong to a family of three AlkB-like Fe(II)/α-ketoglutarate-dependent dioxygenases: TET1, TET2 and TET3 [33]. Several studies discovered that TET proteins can catalyze the successive oxidation of 5mC to 5-hydroxymethylcytosine (5hmC), 5-formylcytosine (5fC) and 5-carboxylcytosine (5caC) [34,35,36]. These modified 5mC products were believed to serve as intermediates for the subsequent conversion back to unmodified cytosine through further reactions [29]. This TET-mediated reaction is found to be dependent on α-ketoglutarate (αKG) which is a key metabolite in TCA Cycle. TETs use oxygen as the substrate for TETs to catalyze the oxidative carboxylation of αKG, producing CO_2_, enzyme-bound succinate and a reactive high-valent Fe(IV-oxo) intermediates [37]. This intermediate then reacts with 5mC/5hmC/5fC for the production of 5hmC/5fC/5caC. Since αKG is produced mainly from oxidative decarboxylation of isocitrate mediated by isocitrate dehydrogenase (IDH) in the TCA cycle, the need of αKG in this process has linked metabolism with active DNA demethylation [27].

## 3. Metabolic-Chromatin Signaling in Different Physiological Contexts

A key question regarding the metabolic connection of gene control is whether changes in metabolite fluxes and concentrations could influence specific gene programs and ultimately cell function or differentiation. Examples of such mechanisms have been shown in different contexts including stem cells differentiation, regulation of inflammation and tumorigenesis. The aim of this review is not to cover extensively these areas but to illustrate examples where metabolic signaling plays a role. The interface of metabolism and gene regulation in the context of metabolic disorders will be discussed.

### 3.1. Metabolic Reprogramming in Stem Cells

Stem cells undergo a metabolic reprogramming to allow cell differentiation [38]. Recent hypothesis suggests that metabolic changes could drive epigenetic modulation of the earliest steps of development. Pluripotent stem cells are able to self-renew and differentiate into all adult tissues in mammals [38]. There are two defined states of pluripotency, “naïve” and “primed” which have higher and reduced developmental potential respectively [39]. Naïve versus primed states have distinct epigenetic states, whereby primed state is more associated with repressive histone methylating marks. Interestingly, the laboratory culture conditions of naïve stem cells require a GSK3β and a MEK inhibitor (known as 2i) which leads to high levels of αKG [40]. The family of DNA and histone demethylating enzymes harbor a dioxygenase catalytic activity, which requires αKG/Fe^2+^. It has been shown that high αKG levels are associated with a demethylating activity which erases DNA and chromatin repressing marks and maintains stem cell pluripotency [41].

Recent work shows that metabolome regulates naïve to primed stated embryonic stem cell transition through the regulation of global H3K27me3 [42]. The authors found that H3K27me3 are indirectly regulated by the activity of nicotinamide *N*-methyltransferase (NNMT), which consumes SAM. NNMT was found particularly elevated during the naïve state which results in decreased SAM levels leading to reduced H3K27me3 [42]. These results contrast however with other findings showing that H3K27me3 is less sensitive to SAM compared to other histone methylations such as H3K4me3 [43,44]. On the other hand, the naïve state requires high levels of histone acetylation, which leads to an open chromatin conformation [39]. In addition, the transition into a primed state is accompanied by a metabolic shift to a high glycolytic status and reduced fatty acid oxidation [42,45]. This leads to high levels of acetyl-CoA production which is used for histone acetylation necessary for pluripotency [46]. On the other hand, differentiation of adipocytes require high levels of acetyl-CoA [6], although this could be also due to the requirements acetyl-CoA for fatty acid synthesis during adipogenesis.

A glycolytic switch also takes places in satellite muscle cells during its activation which leads to proliferation and differentiation into myoblasts [47]. Ryall et al. have shown that this glycolytic shift decreases intracellular NAD^+^, reducing the activity of SIRT1 deacetylase activity [48]. This is associated with an increase in H4K16 acetylation and an activation of the myogenic differentiation program of muscle stem cells [48].

### 3.2. Metabolic Reprogramming in Immune Cells

Recent studies highlight the relevance of metabolic inputs into inflammatory functions by modulating cell differentiation [49]. It has been shown that some metabolites play a signalling function during immune stimulation in both adaptive and innate immunity. This has been studied in the metabolic role of lymphocyte T cell activation and macrophage polarization.

Macrophages are the first line of defense against pathogens. The activation of monocytes into distinct and specialized macrophage polarization states combats bacterial or parasitic infections, namely pro-inflammatory versus anti-inflammatory respectively. Macrophages with anti-inflammatory activation also participate in processes such as wound healing. Both anti- versus pro-inflammatory have profound metabolic differences [50]. The first evidence of a metabolic change during macrophage activation (pro-inflammatory state) dates from the 70s, where a switch from oxidative phosphorylation to glycolysis was identified [51]. The Pro-Inflammatory Toll-like receptor 4 (TLR4) activation by lipopolysaccharide (LPS) results in a shift to glycolytic metabolism and impaired mitochondrial respiration. Furthermore, TLR signaling results in marked shifts in NAD^+^/NADH ratios, which influence the activities of the sirtuins, potentially modifying histone acetylation status [52]. In addition, some evidence suggests that certain metabolites could directly influence macrophage polarization. Liu et al. showed that αKG production via glutaminolysis promotes anti-inflammatory polarization by modulating the histone demethylase JMJD3, which mediates epigenetic changes. In addition, αKG inhibits pro-inflammatory induction through inhibition of NF-κβ [53].

Adaptive immunity seems also to be subjected to metabolic regulation. Activation of lymphocyte T-cells is initiated by the presentation of antigens through the T cell receptor (TCR) by antigen presenting cells (APCs). Naïve CD4^+^ T-cells then undergo differentiation into an activated “effector” state which mediates an immune response. Interestingly, this transition is associated with a metabolic switch from high rates of fatty acid oxidation to aerobic glycolysis [54]. Overall, the metabolic switch sustains the rapid cell proliferation necessary to activate inflammation. Aerobic glycolysis, mediated by increased lactate dehydrogenase activity, induces T helper 1 cell differentiation through elevation of acetyl-CoA levels resulting in histone acetylation of the IFN-γ promoter [55].

Similar to macrophages, αKG metabolism also determines T cell activation through its regulatory role of TET enzymes. A competitive inhibitor of αKG, 2-Hydroxyglutarate (HG) has been found to control histone and DNA methylation by affecting TET enzymes. HG accumulates in CD8^+^ T cells followed T-cell receptor activation and determines T-lymphocyte fate through an epigenetic mechanism [56]. One-carbon metabolism pathway is also important for T cell differentiation, as a dietary restriction of serine and glycine was associated with deficient effector T cell activation and IFN-γ production [54,57].

## 4. Metabolic Signaling in the Regulation of Gene Expression of Metabolic Disorders

Modern lifestyle, particularly in wealthier countries is associated with an increase of sedentarism and excessive food intake. This has led to a triplication of obesity rates within the last 20 years and the trend continues as childhood obesity has also risen [58]. Moreover, obesity is the cause of several associated pathologies including mainly type 2 diabetes (T2D), cardiovascular disease (CVD) and some types of cancer [59,60]. Besides the socio-economic and environmental causes of obesity, the biological mechanisms underlying the regulation of energy balance are still intriguing. Although progress has been made in the understanding of the physiological, endocrine, cellular and molecular functions of different metabolic tissues and the central nervous system, there are still no current effective therapies to treat obesity [58]. Novel alternative therapeutic strategies are needed and advances have been made in particular in the identification of the molecular mechanisms that control energy expenditure through adipose tissue thermogenesis [61]. However, how nutrients and derived intermediary metabolites control energy balance through gene expression regulation is not fully understood.

Given the connection between key intermediary metabolites and chromatin or DNA modifications, a relevant question is how an obesogenic nutrient load would impact the metabolite-chromatin regulatory axis. Because glucose is a major source of acetyl-CoA, it is tempting to speculate that high glucose levels induced by obesity would impact into histone acetylation through modulation of acetyl-CoA metabolism. In relation to this hypothesis, a comprehensive targeted metabolomic approach to quantify acyl-CoAs thioester compounds using the liver of high-fat diet fed mice did not show significant differences on the levels of acetyl-CoA in response to diet [62]. However, Carrer et al. showed a decreased acetyl-CoA and acetyl-CoA/CoA ratio in liver, adipose tissue, and pancreas of mice fed a high-fat diet [63]. This was in addition associated with a global decrease in histone acetylation only in white adipose tissues but not in the liver. This reduction could be resulted from a decrease in the expression of ACLY induced by high-fat diet [63]. It was shown that acetylation of many histone lysines were correlated with acetyl-CoA levels [63]. On the other hand, it is not known whether other histone acylation modifications could be induced upon high-fat diet or exacerbated fatty acid metabolism. Acyl-CoAs can be generated by various intermediate metabolic pathways and it has been shown that short acyl-chains including propionyl-CoA, butyryl-CoA or crotonyl-CoA can post-translationally modify histones [64]. It has been postulated by histone acylation mark actively transcribed genes [64]. In relation to fatty acid metabolism, a recent study using isotope tracing has shown that lipids are also a source of histone acetylation through acetyl-CoA [65]. The authors showed that at least, the short chain lipid octanoate leads to a specific lipid gene program regulation in hepatocytes [65]. However, one of the technical limitations is to unveil whether the acetyl-CoA derived from lipid oxidation is the source of the histone acetylation of the genes whose expression was regulated.

Another interesting question is how different metabolic fluxes would impact on cell differentiation of metabolic tissues, particularly adipocytes, which can be very relevant in the context of obesity. Pioneering work by Wellen et al. showed that in the presence of glucose, ACLY, which is responsible for the conversion of citrate into acetyl-CoA, promotes histone acetylation, including the promoter of *Slc2a4* encoding the glucose transporter GLUT4 [6]. Moreover, ACLY function was also required for adipocyte differentiation, suggesting that acetyl-CoA and high levels of histone acetylation is needed for adipocyte differentiation as we previously described. However, further research is needed to elucidate the dependence of metabolite signaling into specific adipogenic differentiation programs.

Several transcriptional regulators control white and brown adipocyte differentiation, including some brown adipose-specific regulators such as PRDM16, PGC-1α or EBF2 [66]. By means of genetic loss of function, Yang et al. recently showed that AMPK activation leads to the elevation of αKG which promotes the demethylation of the promoter of *Prdm16* by TET enzymes [67]. The authors showed that the specific *Prdm16* demethylation committed pre-adipocytes precursors into brown adipogenic differentiation. Interestingly, pharmacological activation of AMPK through metformin (the main drug used in T2D) or AICAR rescued the obesity-induced suppression of brown adipogenesis and thermogenesis [67].

Histone demethylases have also been involved in the formation of obese phenotypes. For example, the knockout mouse model of the H3K9-specific demethylase KDM3A (also known as JHDM2A or JMJD1) leads to adipose tissue accumulation and insulin resistance [68]. KDM3A was shown to deacetylate H3K9me2 at the peroxisome proliferator activator receptor response element (PPRE) which controls the expression of the thermogenic gene Uncoupling Protein 1 (*Ucp1*) in brown adipose tissue. The expression of KDM3A was in addition induced by environmental cold exposure and recruited PPARγ-RXRα and PGC-1α to the *Ucp1* promoter which enhanced *Ucp1* expression [68]. Recent work has shown that KDM3A is phosphorylated in response to cold-induced adrenergic signaling at S265 [69,70]. Using KDM3A-S265A knock-in mice, the authors showed that S265 phosphorylation is required to the demethylate the promoter of *Ucp1* in beige adipocytes [69,70]. This promoter is highly methylated (H3K9me2) in white adipocytes unless there is a cold-induced chromatin reprogramming leading to KDM3A-dependent demethylation. KDM3A was shown in addition to interact with PRDM16 and recruit the PPARγ-PGC-1α complex [69,70]. Another demethylase, LSD1 was recently showed to mediate repression of white adipose selective targets in brown adipose tissue through the demethylation of the activating mark H3K4me3 [71]. Adipose-specific ablation of LSD1 reduced whole-body energy expenditure through a reduced mitochondrial fatty acid oxidation of the brown adipose tissue which lead to increased fat deposition [71]. Several other chromatin modifying enzymes, particularly HDACs, control multiple metabolic processes. This topic has been extensively reviewed previously [16,72].

Besides the direct function played by chromatin regulators on metabolic processes, how metabolic pathways connect directly the activity of transcription or chromatin factors remains less understood. One example of such regulation is the activation of the transcriptional co-activator PGC-1α. PGC-1α was identified as a cold response inducible factor in brown adipose tissue where it activates thermogenesis through the co-activation of nuclear hormone receptors PPARγ (Peroxisome Proliferator Activator Receptor) [73]. PGC-1α has been shown to connect multiple energy homeostasis pathways through the co-activation of several transcription factors including PPARα, RAR (retinoic acid receptor), and TR (thyroid receptor), ERR (estrogen-related receptor) or YY1 (Yin Yang 1) [74]. The activity of PGC-1α is regulated by its acetylation, PGC-1α is active when it is deacetylated by SIRT1 and deactivated when it is acetylated by GCN5 [75]. Therefore, the acetylation and its dependence on SIRT1 and GCN5 links PGC-1α activity to the nutritional status. A low nutritional status leads to an increase in NAD^+^ which activates SIRT1-dependent PGC-1α deacetylation to activate the transcription factors of energy generating pathways [75]. On the other hand, upon high energy status GCN5 acetylates and represses PGC-1α activity [75]. This has implications in metabolic disorders such as diabetes, since PGC-1α target genes in oxidative phosphorylation were found downregulated in human diabetes [76]. More recently, HDAC3 was shown to also deacetylate PGC-1α in brown adipose tissue and predispose brown adipose tissue to acute cold exposure [77]. 

In addition to chromatin modifying enzymes which sense metabolic status, it has been shown that an increasing number of cytosolic metabolic enzymes is found in the nucleus [78]. They sometimes play additional enzymatic functions which are referred as a moonlight role [78]. Most of cytosolic glycolytic and TCA cycle enzymes are surprisingly also located in the nucleus where they seem to perform functions in transcription or DNA replication and repair or even unknown roles [78]. It is not known how the nuclear translocation of metabolic enzymes is regulated or whether it differs according to the tissue specificity. Post-translational modifications may control the localization of metabolic enzymes in the nucleus, but evidence is lacking. A recent study identified the interaction of the subunits E1b and E2 of the pyruvate dehydrogenase complex (PDH) with the transcription factor STAT5 in the nucleus of adipocytes [79]. Moreover, E1b and E2 were found to be associated with chromatin through STAT5 interaction. Since STAT5 is involved in adipocyte differentiation, the authors suggested that the interaction with PDH subunits could be involved in histone acetylation of STAT5 targets [79].

In summary, future research may reveal additional metabolite-dependent orchestration of complex cellular outputs through chromatin modifications in the context of metabolic disorders.

## 5. Therapeutic Diet Interventions Targeting Metabolic-Chromatin Axis

During the last 200 years, humanity has experienced a doubling of the life expectancy in most developed countries [80]. Scientific advances (immunization against infectious diseases, and antibiotics) together with social (food and water quality, housing and lifestyle) changes are among the main causes [80]. However, we are today facing a burden of non-communicable diseases, usually appearing at late stages in life, including cancer, metabolic disorders and neurodegenerative diseases which reduce the health-span expectancy [80]. Cardiovascular disease and diabetes (mainly due to obesity) are the main causes of global morbidity in both men and women [80].

Diet plays a major role in the development or prevention of diseases, particularly CVD and T2D, however, there are still many open questions partly due to the individual variation and different response to interventions [81,82]. Drug therapies are usually not efficacious, accompanied with side-effects and in most of the cases require a chronic treatment. There is, therefore, an urge to identify novel biological mechanisms and pathways that would lay the ground for novel effective therapeutic approaches or diet interventions.

From a metabolic standpoint, it is tempting to ask how would diet influence on specific gene programs. Particularly, a recurrent question is the effect of nutrition/diet with stable chromatin/DNA modifications. A paradigmatic biological example of the effect of nutrition in epigenetics is the feeding of honeybees (*Apis mellifera*) with “royal jelly”. The feeding of this unique diet to bee larva induces profound phenotypic changes including fertility through its transformation into a “queen” bee. One of the intriguing questions was the fact that, despite the massive phenotypic transformation of the fertile queen, compared to the sterile worker bees, all larvae are genetically identical. The biological mechanism of this transformation, lies on the differential epigenetic DNA methylation, proving a direct link of diet and epigenetics [83]. Potentially, analogous molecular mechanisms may exist in mammals including humans.

Today, ketogenic diet, intermittent fasting or caloric restriction are interventions which have proved some success in some contexts including neurodegenerative diseases, cancer, aging, metabolic disorders or exercise performance [84]. We will address here the type of therapies or interventions which directly interact with chromatin function (Figure 2).

### 5.1. Ketogenesis

A high-fat, adequate protein and very low carbohydrate diet known as ketogenic diet (KD) induce a switch into fatty acid oxidation as fuel usage, resulting in an excessive acetyl-CoA production which leads to ketone bodies formation. In recent years, KD has been used in therapy of epilepsy, is considered metabolically healthy and promotes weight loss. The end product of ketogenesis is β-hydroxybutyrate (βOHB) and its circulating levels have been shown to increase by KD [85]. βOHB has been shown to inhibit class I HDACs and concomitantly increase histone acetylation (H3K9ac and H3K14ac) [86]. Therefore, KD could potentially directly modify chromatin function as some studies suggest.

Daily metabolic fluctuations and circadian rhythms are interlocked, as such; ketogenesis is induced during the daily fasting periods coupled with the circadian clock. KD has been shown to induce circadian-like changes in the gut and liver which were distinctly controlled in both tissues [87]. Particularly, the observed circadian oscillation of serum βOHB induced a coupled cyclic HDAC inhibition and H3 acetylation [87]. Another recent study, found β-hydroxybutyrylation as a novel type of histone modification which was induced in the liver of long-fasted mice [88]. The genes marked by this chromatin mark correlated with the upregulated genes during prolonged fasting [88].

### 5.2. Calorie Restriction

There is a vast literature, including controversies, regarding the potentially beneficial health effects and lifespan extension induced by calorie restriction (CR) in different organisms [89,90]. It is however well supported that the physiological effects of CR involve the activation of sirtuins (NAD^+^ dependent deacetylases) [12,91]. SIRT1 to SIRT6 target different tissues relevant to CR in a systemic coordinated response. These include hypothalamus, skeletal muscle, vasculature, liver, kidney, pancreatic β-cells and adipose tissue [91]. Many of the sirtuin actions in these tissues involve the regulation of key transcriptional regulators. There is a direct link between dietary inputs and sirtuins which play a nutrient-sensing regulatory role. The pharmacological intervention to activate sirtuins has been also intensely pursued as a promising strategy not absent of difficulties. On the other hand, manipulation of NAD^+^ levels was found to ameliorate metabolic and aging dependent disorders [92,93,94,95]. Recently, cellular senescence, a hallmark of aging, has been shown to be delayed by restoring the normal age decline of NAD^+^ through nicotinamide riboside nutrition in mice [96].

## 6. Concluding Remarks

The identification of nutrient regulation of gene expression in the 1960′s set the ground for more complex understanding of how genes are activated or repressed. Then, the era of modern molecular biology has led to a deep knowledge in how cells coordinate transcriptional programs. Nevertheless, how metabolic networks directly confluence with gene expression regulation is still not fully understood. In mammalian systems, hormonal control followed by intracellular signaling cascades are used to sense and transmit environmental signals to regulate specific gene expression programs. Increasing evidence suggests that an additional layer of regulation could be supported by an interface between metabolic intermediates and chromatin modifying enzymes. Metabolites provide therefore a direct nutritional information into gene expression control. Several examples in stem cell differentiation, macrophage or T cell activation and tumorigenesis highlight how specific cell decisions are controlled by the concentration of some key metabolites. Yet, additional examples arise in different fields including metabolic homeostasis where nutritional status plays even a more crucial role in regulating transcriptional outputs. Future research will unveil novel players in the metabolic-chromatin axis which may provide new potential therapeutic targets or specific diet interventions to combat the still expanding rates of obesity and metabolic diseases.

## Figures and Tables

**Figure 1 ijms-19-04108-f001:**
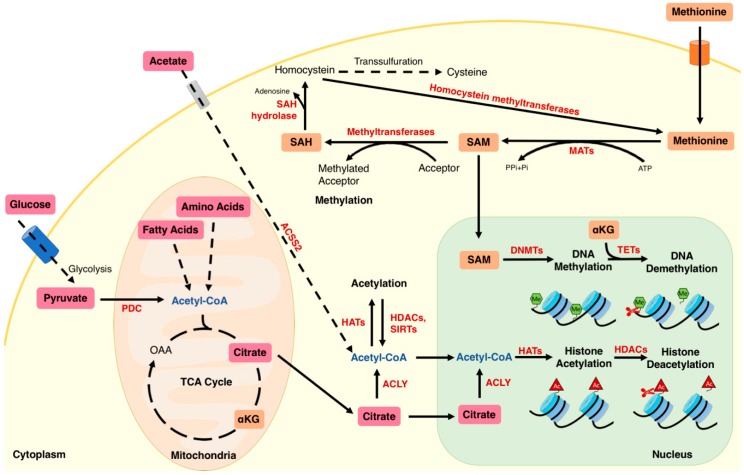
Interaction between metabolism and histone acetylation and DNA/histone methylation. Different nutrient substrates including glucose, fatty acids, amino acids and acetate lead to production of intermediary metabolites which play a role in protein acetylation. Acetyl-CoA derived from glucose, fatty acid or amino acid metabolism is the substrate for histone acetylation after conversion into citrate by TCA cycle and back to Acetyl-CoA in the cytoplasm by ACLY. Acetate is also a source of acetyl-CoA which leads to histone acetylation. Histone and DNA methylation depends on the dietary methionine which enters a cycle for conversion into SAM which is used as a donor of the methyl group. This leads to formation of SAH which is recycled back to methionine through methylation of homocysteine. PDC: Pyruvate Dehydrogenase Complex; ACLY: ATP-dependent Citrate Lyase; SAM: S-Adenosylmethionine; SAH: S-Adenosyl-Homocysteine; DNMTs, DNA N-Methyltransferase; MATs: Methionine Adenosyltransferase. Dashed arrows: multiple-step metabolic pathway; solid arrows: one-step metabolic reaction.

**Figure 2 ijms-19-04108-f002:**
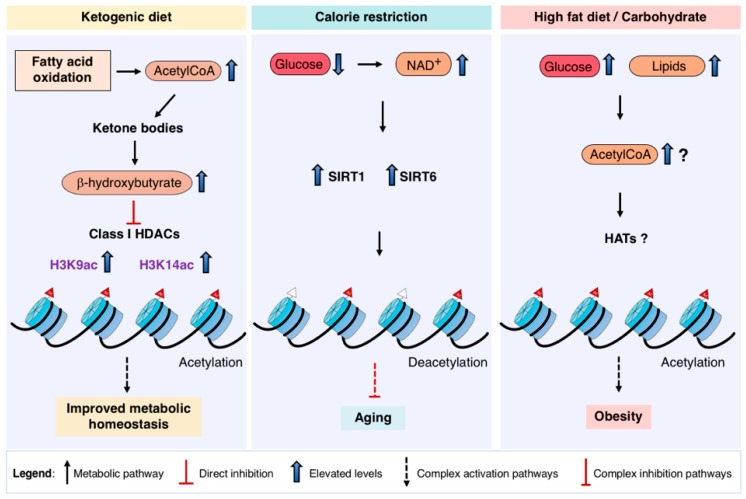
Influence of different diets in chromatin function. Ketogenic diet promotes increased fatty acid oxidation rates which elevates acetyl-CoA production and therefore ketone bodies. The ketone body β-hydroxybutyrate inhibits class I HDACs leading to increased H3K9ac and H3K14ac. Calorie restriction leads to increased NAD+ levels and activation of SIRT1 and SIRT6, which promote histone deacetylation and delays aging. Nutrient overload leads to obesity, it is not fully understood how acetyl-CoA pools may affect specific gene programs in the context of obesity. HDACS: Histone Deacetylases; HATS: Histone Acetyltransferases.

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
