# Peer review of "Metabolic Signaling into Chromatin Modifications in the Regulation of Gene Expression"

_ijms, 2018, doi:10.3390/ijms19124108_

Reviewer 1 Report

The authors have done a good job of describing the roles of metabolic enzymes in regulation of transcription and histone modifications.

The authors are encouraged to add a section on other acylations of histone tails besides acetylation and their roles in regulating transcription. Also, the authors should add a section on roles of H3T11Phos in cancers and how H3t11Phos regulates H3K4me3.

Author Response

See attached pdf. 

Reviewer 2 Report

The review by Gaol’s, Diaz-Hirashi and Verdeguer is very timely, as metabolic control of epigenetic reactions is becoming a very much studied subject.The review is written very well and it extensively covers the current literature. I have only a few remarks to the 2nd part of the manuscript (‘Interaction of metabolites with chromatin function’). Except for the paragraph on acetylation, I am missing the examples of direct connection between metabolites and chromatin / DNA modifications. For instance, how does NAD/NADH ratio impact on histone deacetylation? Also, the authors mentioned that SAM/SAH ration is crucial for determining the catalytic efficiency of methyltransferases, but I would also mention this earlier in the paragraph, when SAH is mentioned for the first time.

Minor point:

p. 6, lines 5,6: ‘ When glutamine, …’ - this sentence does not make sense

Author Response

See attached pdf. 

Reviewer 3 Report

The review describes the influence of metabolism on chromatin organization/gene expression. The topic has been extensively covered by several reviews in the past five years. It is rapidly evolving, so there is room for updates. The review is informative, and displays interesting insights: I found particularly intriguing the review on dietary inputs and the link with metabolic syndrome, which to my knowledge has not been specifically reviewed anywhere else. 

However, I felt the authors wanted to cover a massive topic but eventually missing focus. I could not grasp the biological aspect they wanted to address. Toward the end, the authors seem to focus on metabolic signaling to the epigenome in conditions of metabolic unbalance (such as metabolic syndrome or overnutrition) but I am not sure they got to the point. I actually found this angle very intriguing and potentially innovative as it has never been carefully reviewed before. 

As for now, the manuscript touches dozens of aspects, each one of those have been reviewed in the past. For this reason, the manuscript does not the depth that other, more-focused, reviews have. My recommendation would be to focus more on a specific angle, or alternatively, to limit to one single chromatin modification (acetylation vs methylation).

Please, do that starting with the title, which is confusing in its current state. First: I am not sure "chromatin function" has ever been addressed in the manuscript. Maybe the authors referred to "chromatin organization" or just "epigenome". Second: it is intuitive to think that alterations of the epigenome may lead to changes in gene expression, but the authors never really discuss that in the manuscript. Authors exhaustively present evidence for epigenetic reprogramming in response to diverse metabolic inputs, but how this ultimately impacts gene expression rarely discussed and I don't think the authors ever discuss actual pieces of data about that.

Also the abstract is a little misleading: the lead is that hormonal tissue cross-talk regulates cell metabolism. The authors then build on the fact that chromatin modifications are regulated by metabolite availability. One assumes that the review will discuss how hormones/hormonal imbalance would impact the epigenome, via rewiring of cellular metabolism. That is however never addressed. I think it is actually an excellent point, which could be exploited in a revised version.

The authors also seem to have a loose definition of "physiological contexts": in the intro, metabolic disorders appear to fit in the category, which is not correct. In Chapter 3, cancer is included in physiological contexts, which is also not correct. In this respect, the paragraph on cancer seems to be particularly scant. For the purpose of this review (and considering the body of work presented elsewhere), I would not discuss the metabolic-chromatin axis in cancer.

Please note: the consensus for acetyl-CoA spelling has a dash between acetyl and CoA.

Chapter 2

Page 2, row 1-4: roles of acetyl-CoA are several others, for example in protein glycosilation and synthesis of polyamines. Also, other acetyl-CoA-generating aminoacids exist. Either the authors lay everything out or acknowledge the existence of non-discussed pathways.

Page 2, row 6: acetyl-CoA is the ONLY donor (at least in mammals).

Page 2, row 9: I am not sure there are definitive data that support the fact that acetyl-CoA is produced mainly in the mitochondria. If anything, recent data seem to indicate the opposite (Chen et al, Cell, 2016).

Page 2, row 16: the term "high metabolic status" is not well defined.

Page 2, row 25: significant evidence suggests HATs activity depends also on acetyl-CoA:CoA ratio (Pietrocola, 201%). Probably worth pointing that out.

Page 3, row 14-16: I think others have corroborated those findings (ex: Katsyuba et al, nature, 2018).

Page 3, row 31: I am not sure we can say "immediately undergoes hydrolysis". Or cite reference for kinetics.

Page 3, row 39: missing reference. Also, not sure this is accurate.

Page 3, row 42-45: authors cite a very old paper. There is more recent evidence for metabolic-dependent regulation of histone methylation via SAM availability. Ex: Shyh-Chang et al, Science, 2013

Page 4, row 20: missing reference.

Chapter 3

Page 4, row 30: citation not appropriate

Page 4, row 37: citation missing

Page 4, row 41-44: the sentence is awkwardly phrased. Could the authors make it clearer? Also, I invite to caution, as multiple other studies show that H3K27me3 is relatively insensitive to SAM availability, which seems to have a greater impact on H3K4me3.

Page 4, row 45-49: authors strongly claim that elevated acetyl-CoA is required for pluripotency. Although I generally believe this is true, notable exemptions must be pointed out: for example, elevated acetyl-CoA is important for adipocytes differentiation (Wellen et al, Science, 2009). Please, use caution and note context dependency.  

Page 5, row 6: authors discuss role of metabolic reprogramming in immune cells, not in inflammation.

Page 5, row 8: citation missing

Page 5, row 11: phenotypical polarization of macrophages must be introduced for ease of reading

Page 5, row 18: Sirtuins, not HDACs

Page 5, row 20-25: impact on the epigenome is highly speculative, does not fit the scope of the review

Paragraph 3.3: as mentioned before, either omit or significantly re-write.

Chapter 4

The title misses a final g in signaling

Overall, the chapter might be insightful, but seems to lack focus, and often digresses from the topic outlined in the title. Please, discuss only data actually relevant to obesity/overnutrition/metabolic syndrome. I recommend significant revision.

Page 6, row 32-35: authors speculate on experiments that have actually been made. Carrer et al, JBC, 2017 and Liu et al, Mol Cell Proteomics, 2015

Page 7, row 16-18: this paragraph is really difficult to read. Please, rephrase.

Page 7, row 19-27: not sure a thermogenic defect is a good fit in the Chapter.

Page 7, row 28: in the paragraph 4.2 the authors do not discuss anything relevant to metabolic disorders, but rather present evidence for transgenerational inheritance of metabolic-induced epigenetic reprogramming. Interesting point, but off-topic. Please, adhere to what indicated in the title. Significant re-structuring/re-writing is required.

Page 8, row 3: the paragraph 4.3 might be expanded in an independent Chapter.

Page 8, row 11-13: citation missing

Figures are well designed, but never cited in the text. Moreover, in Figure 2 (KD panel), it seems that the boost in histone acetylation suppresses obesity. I don't think this is demonstrated.

Author Response

see attached pdf

Reviewer 4 Report

I have read the manuscript „Metabolic Signaling to Chromatin Function in the Regulation of Gene Expression“ with the greatest interest.

This is a great manuscript.

The authors gave valuable functional links between metablic signalling and chromatin function.

I like the composition of the paper which allows easy reading. It also seems to me that the authors dedicated exactly enough „space“ to all aspects of the topic they planned to cover. Of course, one can always include more - but I think that "the taste" is just right.

However, it seems to me that the importance of IDH mutations in relation to a distinct DNA methylation phenotype and an altered metabolic profile in glioma must be mentioned. Further, I think that a short paragraph dedicated to the functional connections between epigenetic effects of IDH1/2 mutations and synthetic lethality (downregulation of NRF2 and increased response to  chemotherapy;  downregulation of NAPRT1 and sensitization to NAMPT inhibition; the decrease of BCAT1 and increased dependency on glutaminolysis; increased hydroxyglutarate and its suppressive effect on homologous recombination associated with increased sensitivity to  PARP inhibitors), especially in the paragraph related to cancer, should be mentioned.

Author Response

See attached pdf. 

Round  2

Reviewer 3 Report

I appreciate authors' effort to paling the manuscript to reviewer's comments.